# Thread-Based Modeling and Analysis in Multi-Core-Based V2X Communication Device

**Won-Seok Choi** [1] and **Seong-Gon Choi** [2,*]

1 Research Institute for Computer and Information Communication (RICIC), Chungbuk National University, Chungdae-ro 1, Seowon-Gu, Cheongju 28644, Korea; wschoi@cbnu.ac.kr
2 College of Electrical and Computer Engineering, Chungbuk National University, Chungdae-ro 1, Seowon-Gu, Cheongju 28644, Korea
* Correspondence: choisg@cbnu.ac.kr

**Abstract:** We propose a thread-based modeling and analysis method for a vehicle-to-everything (V2X) communication device, according to the performance requirements of the V2X application. For a developer to develop a V2X device or application, its performance requirements must first be determined. Furthermore, the developer selects the hardware system to enable the service to function while satisfying the performance requirements. Through this process, the developer designs and creates the program in a manner that considers thread configuration and core mapping with the given hardware resources. Then, the performance evaluation of a system is tested via instruction-level program analysis with executable programs. This process is an essential procedure for developing a program that satisfies the performance requirements. However, the debugging procedure repeated through program development, performance analysis, and program modification requires significant time and is costly. Hence, to reduce the time and cost of unnecessary work, we propose a thread-level modeling and analysis method using a queueing theory for a V2X communication device that can be applied at the design level. First, we propose a thread-level performance modeling and analysis method based on queuing theory in a multi-core-based vehicle service system. Furthermore, we analyze the performance of a multi-core-based vehicle service system utilizing the proposed method.

**Keywords:** V2X; thread-level modeling; queueing theory; multi-core-based vehicle application; performance evaluation

## 1. Introduction

Vehicle network technology, which is an advanced network technology, has been developing in a direction that considers the safety and convenience of users, such as cooperative driving, infotainment, and autonomous vehicles with artificial intelligence (AI) and communication infrastructure. In addition, the service utilized in the vehicle network requires multiple simultaneous services, considering user safety and convenience [1–7].

In vehicle communication, the vehicle service is divided into in-vehicle communication services and vehicle-to-everything (V2X) according to the subject and target [8–13]. The V2X communication service is a technology in which a vehicle exchanges information with other vehicles and infrastructure, such as road-side units utilizing wired and wireless networks. It is divided into autonomous/cooperative driving, traffic safety, traffic efficiency, and infotainment services, according to the service requirements. In-vehicle communication services are utilized for in-vehicle control [12–14].

Recently, the in-vehicle communication device has been evolving as an integrated device. In the in-vehicle communication, there is a problem that the complexity of the electronic control units (ECU) increases to about 150 per vehicle depending on the type and level of advanced driver assistance systems technology, so the need for an integrated ECU is emerging [15–20]. Even in V2X communication, if different devices are composed according to the service application, the physical configuration and interworking can

be complicated. Therefore, it needs to be composed as a single integrated device using multiple interfaces [8–11,21–23]. In order for a V2x device to be composed as an integrated device, various network interface technologies considered in V2X communication must be supported, and V2X service requirements must be satisfied. For this, an integrated V2X device supporting various network interfaces based on a multi-core is essential.

To satisfy the service performance requirements of the vehicle service communication device, it is necessary to design a system that considers performance analysis in the development stage. Therefore, to develop a vehicle communication service system, a developer selects a hardware system and performs designing and programming to satisfy the performance requirements of the targeted services.

Figure 1 shows a simplified model of software development based on a waterfall model [24]. This model is a procedure for software development and consists of requirements, design, code and debug, test and pre-operations, and operations and maintenance stages. Each stage is affected by the previous stage and affects the next stage. Therefore, in order to solve a problem at a certain stage, it is necessary to consider solving the problem together with the previous stage, and the result of the solution affects the next stage.

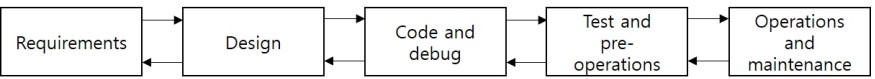

**Figure 1.** Simplified model of software development based on waterfall model [24].

According to the above model shown Figure 1, V2X-communication-related developers develop an embedded device through software design–development–debugging processes that consider hardware selection and hardware resources with the goal of satisfying requirements. After prototype development, the performance is checked by measuring the performance of the embedded device and trying to achieve the target performance by repeating the debugging process of analysis–correction–measurement. In this case, developers usually choose hardware high to achieve the expected performance. This debugging process is repeated until the performance is satisfied. If this method does not address the challenge, the system will be altered to a higher-level hardware specification for performance satisfaction. However, this solution increases the cost of the device, and decreases the system efficiency as the hardware specifications increase. The problem that increases the development cost is caused by inaccurate hardware selection at the design level and performance analysis through the complicated debugging process at the instruction level. Therefore, for the efficient program development of embedded devices, a method that can easily analyze and evaluate performance at the design level is required.

In other words, for efficient program development of V2X communication devices, performance analysis utilizing thread-level modeling is required in the design stage before program development. The modeling method at the thread level can analyze the performance during the program development phase. Therefore, unnecessary processes and costs of the instruction-level program performance analysis in the debugging process can be reduced. Specifically, the performance analysis at the thread level enables detailed design by appropriately configuring the thread, CPU, and memory resources of the V2X communication device, compared to the service requirement, and can reduce unnecessary debugging processes and costs [25–29].

Most of the research related to performance analysis in V2X communication consist in test method studies to verify the functions and performance in the system aspect based on the developed results, such as program, device, and system. These researches study how to configure and measure the test environment for V2X device and system testing depending on each purpose. According to the test purpose, there is the conformance testing method, the function testing method, the performance testing method, the vehicle gateway testing method, the penetration testing method, the accelerated testing method, the field testing method, etc. [30].

The conformance testing method tests the protocol conformance for V2X commutation [31–33]. The function testing method checks the function of the application for

various scenarios to ensure the reliability and efficiency of the V2X application [34–43]. The performance testing method evaluates whether the latency, reliability, throughput, etc., of the V2X application are guaranteed [44–52]. The vehicle gateway testing method checks whether the vehicle gateway is running correctly [53–56]. The penetration testing method tests the security of the target system by simulating the method of a malicious attacker [57–62]. The accelerated testing method testing on a public load is a test method to solve the problem that takes a long time [63,64]. The field testing method provides a method to conduct tests in the testbed before official promotion [65,66].

However, these studies related to performance analysis in the V2X study of the performance measurement method for the verification of functions and the performance in the system aspect based on developed programs. In other words, these studies should perform the debugging process using a complex and expensive instruction-level analysis method to analyze the performance after returning previous stage, as shown in Figure 1.

To address this challenge, we propose a modeling and analysis method according to the required performance of the V2X communication service program via thread-level modeling utilizing the queueing theory of the V2X communication service program. In addition, we analyze a model of a multi-core-based vehicle service system utilizing the proposed method.

The contributions of this study are as follows:

- We describe the architecture of the V2X communication device and its thread-level performance analysis possibility. We also describe the architecture of a multi-core-based V2X communication device. In particular, we analyze the architecture of a V2X communication device with a multi-network interface and a multi-core to support multiple V2X services according to V2X service requirements. We also describe how events such as packets, queries, and transactions occurring in the V2X communication device can be recognized and modeled as workloads at the thread level.
- We propose a thread-level performance analysis method utilizing queueing theory in the design phase of a multi-core-based vehicle service system. We also define a multiple-distributor and multiple-application model to provide various V2X services in the V2X communication device using a multiple-network interface and a multi-core resource. In addition, we model the operation at the thread level and performance indicators such as latency and throughput based on the queuing theory for the multiple-distributor and multiple-application model.
- We analyze the performance of the test models of a multi-core-based vehicle service system utilizing the proposed thread-level performance analysis method. We also analyze the performance of three core allocation types using the V2X communication device model based on queuing theory. Furthermore, we analyze the satisfaction of V2X service requirements according to the allocation of core resources in multi-core-based V2X communication devices.

The remainder of this paper is organized as follows: We first introduce V2X service requirements and testing and measurement methods for the performance analysis of V2X communication in Section 2. In Section 3, we introduce the proposed method in detail. Section 4 reports the modeling for some analysis model and test results, and provides a discussion and analysis. Finally, conclusions and future studies are provided in Section 5.

## 2. Related Work

In this section, we discuss related work on V2X service requirements and the V2X communication testing and performance measurement method.

### 2.1. V2X Service Requirement

Table 1 lists the service requirements for the V2X. V2X communication has a variety of applications and services. Each of these applications and services has different throughput, latency, and frequency requirements. Therefore, these applications and services are often grouped into four categories: autonomous/cooperative driving, traffic safety, traffic efficiency, and entertainment.

**Table 1.** Service requirement of vehicle application.

| Application | Latency | Throughput | Device Density (km²) | Number of Devices per Cell | Communication Rang (m) | % of Mobile Devices | Mobility Speed (km/h) | Traffic Type | WAVE / IEEE 802.11p | IEEE 802.11n/ac | C-V2X |
|---|---|---|---|---|---|---|---|---|---|---|---|
| Autonomous/ cooperative driving | ≤10 ms | ≥5 Mbps | Urban 3000 Highway 500 | Urban 300 Highway 50 | Urban 500 Highway 2000 | >95 | Urban < 100 Highway < 500 | Event-triggered | Urban O Highway X | X | 4G X 5G O |
| Traffic safety | 20–50 ms | 700 Mbps | Urban 3000 Highway 500 | Urban 300 Highway 50 | Urban 500 Highway 2000 | >90 | Urban < 100 Highway < 500 | Event-triggered | X | X | 4G X 5G O |
| Traffic efficiency | 100–500 ms | 10–15 Mbps | 3000 | 300 | 2000 | >80 | <500 | Periodic | X | X | O |
| Infotainment | ≤1 s | Around 80 Mbps | – | – | – | – | – | Real-time | O | O | △ |

Autonomous/cooperative driving mainly focuses on V2X communication between vehicles in close proximity. This particular application has stringent requirements in terms of communication throughput and latency. More specifically, these applications require a throughput of 5 Mbps and a latency of 10 ms [67–70].

Traffic safety represents a more general view of autonomous/cooperative driving applications. Traffic safety applications have several goals, including reducing the number and severity of collisions between vehicles, protecting vulnerable road users, and reducing property damage. As expected, these applications have stringent requirements. For example, the minimum requirement for pre-detected collision warnings is a round-trip latency of 20–50 ms. In addition, the throughput required for traffic safety services such as road sign recognition is estimated to be 700 Mbps [67,68,71–73].

Traffic efficiency focuses on a variety of tasks such as coordinating intersection timing, planning routes from the origin to the destination for various vehicles, and sharing general information, including on geographic location and road conditions. These applications often have less stringent requirements than other applications. For example, the acceptable latency for these applications ranges from to 100–500 ms and throughput ranges from 10–45 Mbps [67,68,72].

Infotainment refers to a service that provides non-driving-related general information (e.g., rental car service location) and entertainment (e.g., video streaming). These applications and services typically have the lowest requirements. For example, a latency of up to 1 s can be achieved. In addition, the throughput requirements are estimated to be approximately 80 Mbps, which is similar to the requirements for conventional mobile services [67,69,73].

One of the important criteria for selecting the network interface technology to be utilized based on these categories is the communication range. Anonymous/cooperative driving, traffic safety, and traffic efficiency services require relatively strict communication rate performance compared to infotainment services to provide seamless service while driving.

The latency, throughput, and communication range in Table 1 become key performance parameters that the developed application must satisfy. According to these parameters, it is necessary to selectively utilize air interface technologies such as Cellular-V2X(C-V2X) and IEEE 802.11.

In Table 1, notations of O, X, and △ show some of the available network interface technologies compared to the service requirements of the four service categories. O means it is acceptable to use for the service. X means it cannot be used for the service. *D* means acceptable but not suitable for cost-effective reasons. WAVE/802.11p is suitable for autonomous/cooperative driving service and infotainment service in urban areas. IEEE 802.11n/ac is suitable for infotainment service, and 4G of C-V2X is suitable for traffic efficiency service. However, to provide autonomous/cooperative driving service with C-V2X, 5G or higher technology is required. Comparing the C-V2X and the IEEE 802.11 series, C-V2X is a network interface technology suitable for overall services. However, C-V2X has a higher communication cost than the IEEE 802.11 series, and it can thus be said that the IEEE 802.11 series is more suitable for infotainment services with a significant amount of traffic.

It is therefore necessary to utilize wireless interface technology in a mixed method according to the required performance of the services utilized and the infrastructure environment for providing the service.

Therefore, to operate the V2X service application while satisfying the performance requirements of the vehicle service communication device, the developer must properly configure the thread and core to satisfy the requirements according to the above service type. Accordingly, we propose a performance analysis method at the thread level to test and analyze whether the above performance requirements are satisfied when a developer develops a V2X service application in a vehicle communication device.

## 2.2. Testing and Measurement Methods in V2X Communication

In this section, we introduce research related to testing and measurement methods for performance analysis in V2X communication.

Figure 2 shows the classification of the performance analysis method of V2X communication. The performance analysis method of V2X communication can be classified into a design-level performance analysis method and a prototype-level performance analysis method. The prototype-level performance analysis method can be classified into service-, device-, and system-level testing [9,31–66] and instruction-level testing [28,29]. The design-level testing is performed at the design stage, shown in Figure 1. The design-level testing is introduced in Section 3 as a proposed method.

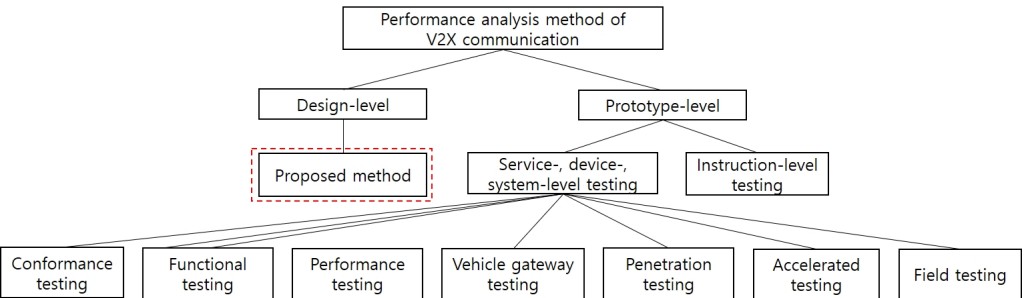

**Figure 2.** Classification of performance analysis method of V2X communication.

Service-, device-, and system-level testing are performed at the testing and pre-operations stages, as shown in Figure 1. According to the test purpose, service-, device-, and system-level testing are a conformance testing method, a function testing method, a performance testing method, a vehicle gateway testing method, a penetration testing method, an accelerated testing method, a field testing method, etc. [30].

Protocol conformance is the basis of V2X communication and tests interoperability between vehicles, pedestrians, road-side unit (RSU), cloud platform, and other participants. As a conformance testing method, ETSI standardized a test specification for V2X conformance testing [31]. ETSI recommends using TTCN-3 to implement the test specification in accordance with ISO 9646. 3GPP provides a test system architecture based on TTCN-3 [32]. The architecture of 3GPP is similar to the ETSI test specification, but there are some differences. The TTCN-3 test system communicates with the device being tested via the system simulator hardware. ISO TS 20026 also described test architectures similar to ETSI and 3GPP [33].

Functional testing determines whether the application can guarantee stability and efficiency to be triggered by V2X scenarios [30]. Aramrattana et al. [34] presented a simulation framework consisting of driving, traffic, and network simulators for testing and evaluation of cooperative intelligent transport system applications. Mittal et al. [35] presented a comparative study of various publicly available VANET simulation tools, namely, GrooveNet, TraNS, and NCTUns. Furthermore, they presented a comparative study of various network simulation tools, namely, NS-2, OPNET, GloMoSim, QualNet, and Vanet-MobiSim. Kim et al. [36] proposed a method of combining a network and traffic simulator for building a test bed using characteristics of a real road environment such as mobility and a channel model called V2XREF. Schiller et al. [37] proposed a virtualization-based framework for emulating vehicular ad hoc networks for evaluating and testing network-aware automotive embedded systems. Choundhury et al. [38] proposed an integrated simulation environment for testing V2X protocols and applications that combines three simulators such as a traffic simulator, a network simulator, and an application simulator. Ahmed et al. [39] proposed a flexible VANET testbed architecture that adopts multistage emulators. Ming et al. [40] proposed a testing framework based on a Veins simulation platform for securing the VANET application and demonstrated its practicability using a proof-of-concept example with attacks of malicious messages in an expressway tolling



application over VANET. Riberio et al. [41] proposed a platooning management protocol that carries out implementation and testing by means of simulation, using the V2X simulation runtime infrastructure framework. Buse [42] proposed a vehicle testing system that combines a VANET simulator with a 3D driving simulator for supporting the development process of next-generation ADAS systems. Szendrei et al. [43] proposed a SUMO-based hardware-in-the-loop V2X simulation framework for the cost-efficient testing and rapid prototyping of cooperative vehicular applications.

Performance testing is an important means for testing and guaranteeing the requirements of V2X applications, such as latency, throughput, etc. [30]. Qin et al. [44] implemented the WAVE protocols and evaluated the latency and packets by comparing the simulation results. Hiromori et al. [45] proposed a protocol testing and performance evaluation method for a set of designated node density distributions and their variations in VANET. Phouthone et al. [46] simulated and analyzed the performance of three routing protocols, namely, DSDV, AODV, and AOMDV for VANET using NS-2 and VanetMobiSim. Marzak et al. [47] analyzed the performance of reactive, proactive, and position routing protocols in comparison with the cluster-based routing protocols AMACAD and MOBIC for end-to-end delay, packet delivery ratio, and bandwidth. Prakash et al. [48] evaluated the performance of IEEE 802.11p by varying the data rate and the node density using NS-3 in VANET. Huang et al. [49] tested the real-world performance of dedicated short-range communication (DSRC) using a large set of naturalistic driving data obtained through the University of Michigan Safety Pilot Model Deployment project. Shi et al. [50] evaluated the performance of DSRC and LTE-V communication based on a typical V2X application at intersection. Furthermore, they built a probability model to evaluate the communication performance under the application Intersection Collision Warning and proposed a multi-hop V2X system with RSUs acting as a relay station of the V2V messages. Kawasaki et al. [51] evaluated the end-to-end latency of PC4-based and Uu-based LTE V2X communication. Nguyen et al. [52] experimented the packet delivery ratio of DSRC safety messages in line-of-sight and non-line-of-sight scenarios. Furthermore, they proposed two relaying strategies, namely, simple relaying and network-coded relaying, by placing an RSU at the intersection center to re-transmit some safety messages to improve the transmission reliability.

Vehicle gateway testing is a means of verifying that the functionality, performance, and security of a vehicle gateway are running correctly [30]. Yang et al. [53] developed an automatic CAN gateway test system to solve the problem of hardship and complexity in manual vehicle gateway tests. Kim et al. [54] proposed a gateway framework for in-vehicle networks based on CAN, FexRay, and Ethernet that would be easy to reuse and verify in order to reduce development costs and time. Lee et al. [55] proposed a high performance for a CAB/FlexRay gateway using a hardware/software co-design method which was implemented with the Xilinx FPGA system. Xu et al. [56] proposed an automatic test system for a vehicle gateway controller based on the CANoe software and the CANcase interface card.

Penetration testing is a method of testing the security of a target system by simulating the tactics of a malicious attacker [30]. Khan [57] proposed a testing method of vehicle network security which provides security validation against requirements, as well as security validation using a white box approach, a black box approach, and gray box approaches. Prathap et al. [58] examined how to perform penetration testing of ECUs by discussing the working of ECUs and its security mechanisms, and discovering the vulnerabilities that exist in the ECU. Furthermore, they suggested appropriate countermeasures to patch the vulnerabilities for the discovered attacks on ECUs. Hakeem et al. [59,60] proposed a lightweight V2X security-protocol-based simulation and a light-weight message authentication and privacy preservation protocol for V2X communications. Moreover, they implemented the proposed protocol using commercial V2X devices to prove its performance advantages over the standard and non-standard protocols. Dürrwang et al. [61] proposed an approach to support penetration testing by combining safety and security analytics to enhance auto-

motive security testing using a reuse method of the results. Kobezak et al. [62] proposed a testing framework of unmanned system penetration.

Accelerated testing methods are being studied to effectively reduce the cost and time required for the vehicle reliability verification process [30]. Zhao et al. [63] presented an accelerated evaluation process which eliminates the many miles of uneventful driving activity to filter out only the potentially dangerous driving situations where an automated vehicle needs to respond, creating a faster, less expensive testing program. Zhao et al. [64] proposed accelerated evaluation, focusing on the car-following scenario. They modified it to generate more intense interactions between human-controlled vehicles and automated vehicles to accelerate the evaluation procedure.

Field testing must be run before the V2X network and its applications are officially used [30]. Weiß [65] described the German field operational test SimTD, which is an operational test to evaluate the effectiveness and benefits of applications based on vehicular communication in a setup that is representative for a realistic deployment environment. Klapez et al. [66] studied the application-level performance of safety-related communications over IEEE 802.11p, presenting the first outcomes resulting from several short- and medium-range field tests. Ma et al. [9] proposed a cooperative autonomous-driving-oriented MEC-aided 5G-V2X prototype system design based on a next-generation radio ace network experimental platform, a cooperative driving vehicle platoon, and an MEC server providing a high-definition 3D dynamic map service. Moreover, they conducted the field testing at ShanghaiTech University.

As above, recent research on performance analysis is focused on how to test and simulate the performance of the vehicle services, the device, or the system after prototype development [9,30–66]. In addition, the performance analysis of programs involves instruction-level program analysis utilizing execution files after programming [25–29]. The instruction-level testing is performed at the code and debug stage, shown in Figure 1. The instruction-level testing is performed through code analysis of low-level or high-level programming languages of executable programs. This is usually performed using a method called program debugging, which takes significant time and is costly.

In addition, when target performance is not obtained in service-, device-, and system-level testing, there is the problem that the instruction-level performance analysis must be performed again to find the cause of performance degradation.

Despite these issues, the manner in which to perform a performance analysis at the design level has not been considered.

Therefore, for the efficient program development of vehicle service communication devices, performance analysis utilizing thread-level modeling is required at the design stage before program development. The modeling method at the thread level can analyze the performance during the program development phase. Therefore, unnecessary processes and costs in the instruction-level program performance analysis can be reduced. Specifically, the performance analysis at the thread level enables detailed design by appropriately configuring the thread, CPU, and memory resources of the V2X communication device, compared to the service requirement, and can reduce unnecessary debugging processes and costs [25–29].

To address this challenge, we propose a modeling and analysis method for the CPU resource requirement, according to the required performance of the V2X communication service program via thread-level modeling utilizing the queueing theory of the V2X communication service program. In addition, we analyze the model of a multi-core-based vehicle service system utilizing the proposed method.

## 3. Proposed Performance Measure Method of V2X Communication Device

In this section, we propose a performance analysis method at the thread level to analyze the performance at the design phase and verify whether the performance requirements are satisfied when a developer develops a V2X service application in a V2X communication device. Accordingly, we explain a considered V2X communication device. Furthermore,

the V2X communication device is modeled, and a performance measurement method is proposed.

### 3.1. V2X Communication Device

We consider a multi-core-based V2X communication device architecture with various network interfaces, as illustrated in Figure 3. This configuration focuses on the characterization of the multi-core functions, multi-threaded functions, and network interface functions common to most communication multi-core processor architectures, and summarizes all other components in a highly abstract form.

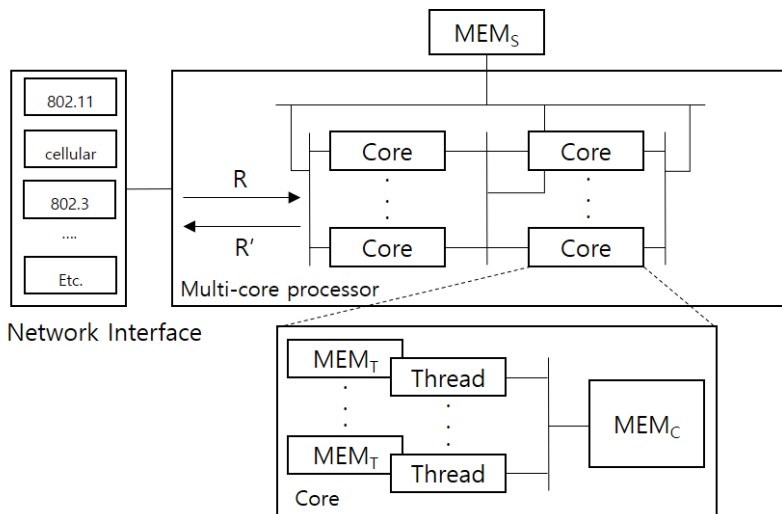

**Figure 3.** Multi-core-based V2X communication device architecture with various network interfaces.

More specifically, in this architecture, a communications processor is generally considered to comprise I/O interfaces, memory, level 1 and level 2 caches, special processing units, scratch pads, embedded general-purpose CPUs, and coprocessors. These supporting components can be represented at three levels: thread, core, and system, collectively referred to as MEMT, MEMC, and MEMS, respectively. Each core can run more than one thread, and the threads are scheduled according to a given thread-scheduling principle.

The network interface may comprise various types of networks, such as IEEE 802.11 series, the LTE network, and IEEE 802.3 interfaces. Packets over the network interface are transmitted to the cores for vehicle service processing. Cores can be configured in parallel and/or multi-stage pipelines, resulting in a stream of packets coming in at one side, at a rate R and out through the other at a rate R′.

The core can be configured as a parallel and/or multi-stage pipeline, resulting in a stream of packets coming in at one side at a rate R, and out through the other at a rate R′.

This packet stream can be either a request (e.g., a database query) or a transaction. For purposes such as packet distributors and vehicle services, packet processing tasks can be split and mapped to different cores in different pipeline stages, or to different cores in a given stage. The dispatcher distributes incoming packets which are temporarily stored in the input buffer to other core pipelines according to a given policy. If the buffer overflows, packet loss can occur.

One of the key ways to measure performance at the thread level is workload modeling. For any application based on a packet, query, transaction, or program, the program task is mapped to a given thread on a given core. The instantiation of the execution of program tasks, which can then be triggered by packets, queries, transactions, or specific input parameters, forms so-called code paths. Regarding a communication processor, a program task mapped to a thread may have an on-and-off characteristic, which is a form of a packet arrival process. The same is true for query/transaction-based applications. The code path that a thread has to process during a given amount of time depends on whether or not there

is actually a mix of different types of packets arriving during that time. Code paths are defined at the thread level, comprising a series of segments corresponding to various events that significantly affect thread-level performance. Based on the pseudo-code snippets for program operations mapped to threads, events can be identified, and their segment length can be estimated [25–27].

Therefore, in this study, the performance of the V2X communication device is analyzed by considering the interaction at the thread level.

### 3.2. Modeling

In terms of per-hop behavior, the throughput and latency of a network interface are only important for V2X communication. However, since the V2X service device includes supportable V2X services, the performance of the network interface, the application, and the packet distributor performing I/O operations between the network interface and the application affects the performance of the V2X service. That is, the performance of the practical V2X service device should consider not only the ability of the network interface but also the performance provided to the distributor and the application. Since the program developer of the V2X communication device uses a commercial module, the performance of the network interface is hardly changeable by the developer. Therefore, we propose a design-level performance analysis method of the V2X communication device based on the thread used by the packet distributor and the application, targeting the core resources of the host system that the developer can access. That is, we propose a thread-based modeling and analysis method in a multi-core-based V2X communication device so that the developer can check the performance when making a programming plan in the design level of the V2X service program.

Figure 4 illustrates the multiple-distributor/multiple application models of the V2X communication device. We consider cores in Figure 3 as the main resource for performance at the thread level. The V2X communication device illustrated in Figure 4 comprises two applications and two packet distributors. Application 1 is a program for providing infotainment services with relatively low latency performance requirements but high throughput performance requirements, and application 2 is a program for providing autonomous/cooperative driving services with relatively high latency performance requirements. The packet distributor transmits packets to the core utilized by applications from network interfaces such as C-V2X and IEEE 802.11 series, or vice versa, to the network. When a packet is input from the vehicle network to the network interface card, the packet is transmitted to application 1 or application 2 through the packet distributor. In addition, when the application needs to transmit a packet to the vehicle network, the application transmits the packet to the packet distributor associated with the network interface card, and the packet distributor transmits the packet to the vehicle network through the network interface card. Applications and packet distributors can allocate multiple CPU cores to satisfy performance requirements.

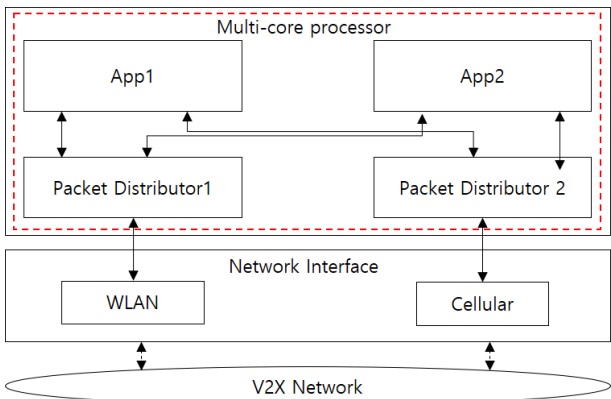

**Figure 4.** Multiple-distributor/multiple-application model of V2X Communication Device.

Figure 5 illustrates a queuing model for the multi-core-based V2X communication device. To facilitate the analysis of the V2X communication device in a steady state, it is modeled as a closed queuing network [74].

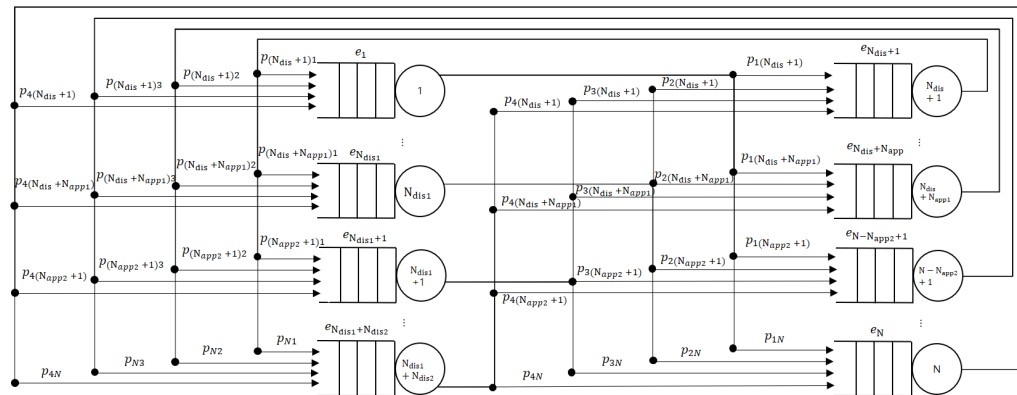

**Figure 5.** Queueing model of multi-core-based V2X communication device.

Packet distributors and applications of multiple-distributor/multiple-application models can allocate multiple cores according to performance requirements. Performance analysis is performed utilizing the number of threads defined by the developer, the number of cores allocated by the developer, and the expected relative arrival rate.

Table 2 presents the notations for queuing modeling of Figure 5. $N_{dis1}$ is the number of cores allocated to packet distributor 1 from core number 1 to core number $N_{dis1}$. $N_{dis2}$ is the number of cores allocated to packet distributor 2 from core number $N_{dis1}+1$ to $N_{dis1}+N_{dis2}$ for packet distributor 2. $N_{app1}$ is the number of cores allocated to application 1 from core number $N_{dis}+1$ to core number $N_{dis}+N_{app1}$. $N_{app2}$ is the number of cores allocated to application 2 from the core number $N - N_{app2} + 1$ to core number $N$. $e_i$ means a relative arrival rate of node $i$. $p_{ij}$ is a routing probability from node $i$ to node $j$. $\alpha$ is a relative arrival rate allocated to cores of packet distributor 1 as follows:

$$\alpha = \sum_{i=1}^{N_{dis1}} e_i, \quad \text{for} \quad 1 \leq i \leq N_{dis1} \tag{1}$$

**Table 2.** Notation.

| Symbol | Meaning |
|:---:|:---:|
| $N$ | Total number of nodes. |
| $N = N_{dis} + N_{app}$ | |
| $N_{dis}$ | Total number of nodes for packet distributor. |
| $N_{dis} = N_{dis1} + N_{dis2}$ | |
| $N_{app}$ | Total number of nodes for application. |
| $N_{app} = N_{app1} + N_{app2}$ | |
| $N_{dis1}$ | Total number of nodes for packet distributor 1. |
| $N_{dis2}$ | Total number of nodes for packet distributor 2. |
| $N_{app1}$ | Total number of nodes for application 1. |
| $N_{app2}$ | Total number of nodes for application 2. |
| $e_i$ | The relative arrival rate of node $i$ |
| $p_{ij}$ | The routing probability to node $j$ from node $i$. |
| $\alpha$ | The relative arrival rate for packet distributor 1. |
| $\beta$ | The relative arrival rate for application 1. |

The relative arrival rate of the cores allocated to packet distributor 2 according to the above equation is as follows:

$$\sum_{i=N_{dis1}+1}^{N_{dis1}+N_{dis2}} e_i = 1 - \alpha, \quad \text{for} \quad N_{dis1} + 1 \leq i \leq N_{dis1} + N_{dis2} \tag{2}$$

$\beta$ is the relative arrival rate of the cores allocated to application 1, as follows:

$$\beta = \sum_{i=N_{dis}+1}^{N_{dis}+N_{app1}} e_i, \quad \text{for} \quad N_{dis} + 1 \leq i \leq N_{dis} + N_{app1} \tag{3}$$

The relative arrival rate of the cores allocated to application 2 according to the above equation is as follows:

$$\sum_{i=N-N_{app2}+1}^{N} e_i = 1 - \beta, \quad \text{for} \quad N - N_{app2} + 1 \leq i \leq N \tag{4}$$

### 3.3. Performance Measures

We introduce a performance measuring method for the V2X communication device. Specifically, we propose a method to measure the performance according to the core resource used by packet distributors and applications of the V2X communication device, not the network interface. For the performance required by the service in the V2X communication device, throughput and latency are critical. Therefore, we model the workload to measure performance at the thread level and derive throughput and latency, which are the main performance indicators of V2X communication devices.

We consider this model to be a closed queueing network for steady-state analysis. For any application based on a packet, query, transaction, or program, the program task is mapped to a given thread on a given core. First, we define $K$ as the total number of threads (or jobs) in a V2X communication system as follows:

$$K = \sum_{i=1}^{N} k_i, \tag{5}$$

where $k_i$ is the number of threads at the node $i$.

According to the Gordon/Newell theorem [74], the state probabilities are given by the following product-form expression:

$$\pi(k_1, \dots, k_N) = \frac{1}{G(K)} \sum_{i=1}^{N} F_i(k_i), \tag{6}$$

where $G(K)$ is the normalization constant or generation function. It is given by the following equation with the condition that the sum of all network state probabilities equals 1:

$$G(K) = \sum_{\sum_{i=1}^{N} k_i = K} \prod_{i=1}^{N} F_i(K_i) \tag{7}$$

The $F_i(k_i)$ is the relative state probability that corresponds to the state probabilities at node $i$ and is defined as follows:

$$F_i(k_i) = \left(\frac{e_i}{\mu_i}\right)^{k_i} \cdot \frac{1}{\gamma_i(k_i)} \tag{8}$$

The visit ratio $e_i$ can be calculated utilizing routing probabilities as follows:

$$e_i = \sum_{i=1}^{N} e_j p_{ji} \tag{9}$$

Furthermore, the function $\gamma_i(k_i)$ is given by:

$$\gamma_i(k_i) = \begin{cases} k_i! & k_i \leq m_i \\ m_i! \cdot m_i^{k_i - m_i} & k_i \geq m_i \\ 1 & m_i = 1, \end{cases} \tag{10}$$

where $m_i$ is the number of servers in node $i$.

Based on the normalization constant defined above, the relevant performance measures can be derived as follows: The marginal probability $\pi_i(k)$ that there are exactly $k_i = k$ threads at node $i$ is given by

$$\pi_i(k) = (\frac{e_i}{\mu_i})^k \cdot \frac{1}{G(K)} \cdot (G(K-k) - \frac{e_i}{\mu_i}) \cdot G(K-k-1)), \quad \text{where} \quad G(k) = 0 \quad \text{for} \quad k < 0 \tag{11}$$

The throughput of node $i$ is given by:

$$\lambda_i(K) = e_i \cdot \frac{G(K-1)}{G(K)} \tag{12}$$

The utilization of a node is given by:

$$\rho_i = \frac{e_i}{m_i \mu i} \cdot \frac{G(K-1)}{G(K)} \tag{13}$$

The mean number of threads for a single server node can be calculated as follows:

$$\overline{K}_i = \sum_{k=1}^{K} (\frac{e_i}{\mu_i})^k \cdot \frac{G(K-k)}{G(K)} \tag{14}$$

According to Little's law [75], the mean response time of threads at node $i$ can be determined as follows:

$$\overline{T}_i = \frac{\overline{K}_i}{\lambda_i} = \sum_{k=1}^{K} (\frac{e_i}{\mu_i})^k \cdot \frac{G(K-k)}{e_i \cdot G(K-1)} \tag{15}$$

The thread-level performance is measured by deriving the relation between threads and throughput and between threads and latency from Equations (12), (14) and (15).

We propose performance measures through modeling as a closed product-form queuing network. Therefore, regardless of the traffic model, performance is analyzed through the expected relative arrival rate, the number of threads, and the number of cores.

## 4. Performance Analysis

In this section, we define three types of queuing models of core allocation for performance analysis, and analyze each model's performance.

### 4.1. Analysis Model

In this paper, we analyze the performance in terms of a V2X communication device to satisfy the performance requirement from an end-to-end perspective for throughput and latency.

Figures 6–8 illustrate three queueing models of core allocation types to test. These models utilize two packet distributors and two applications. We model for different core allocation cases of a V2X communication device as an example.

Figure 6 shows queueing model of core allocation type 1. There are four nodes in a V2X communication device. The core of node 1 is allocated to the packet distributor 1. The core of node 2 is allocated to the packet distributor 2. The core of node 3 is allocated to application 1. The core of node 4 is allocated to application 2.

In the queueing model of core allocation type 1, the relative arrival rates $e_i$ are given as follows:

- $e_1 = \alpha$
- $e_2 = 1 - \alpha$
- $e_3 = \alpha\, p_{13} + (1 - \alpha)\, p_{23} = \beta$
- $e_4 = 1 - \{\alpha\, p_{13} + (1 - \alpha)\, p_{23}\} = \alpha\, p_{23} + (1 - \alpha)\, p_{24} = 1 - \beta$

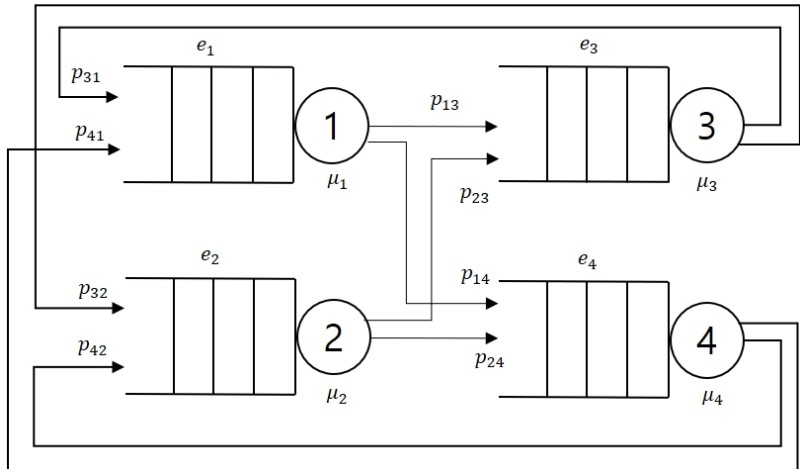

**Figure 6.** Core allocation type 1: queueing model using 4 cores.

Figure 7 shows the queueing model of core allocation type 2. There are five nodes in a V2X communication device. The core of node 1 is allocated to the packet distributor 1. The core of node 2 is allocated to the packet distributor 2. The core of node 3 is allocated to application 1. The cores of nodes 4 and 5 are allocated to application 2.

In the queueing model of core allocation type 2, the relative arrival rates for $e_i$ are given as follows:

- $e_1 = \alpha$
- $e_2 = 1 - \alpha$
- $e_3 = \alpha\, p_{13} + (1 - \alpha)\, p_{23} = \beta$
- $e_4 = \alpha\, p_{14} + (1 - \alpha)\, p_{24} = (1 - \beta)/2$
- $e_5 = \alpha\, p_{15} + (1 - \alpha)\, p_{25} = (1 - \beta)/2$

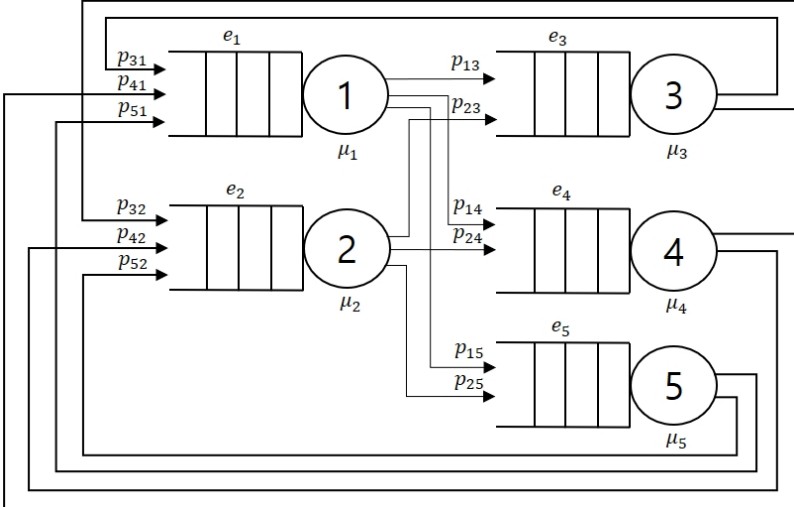

**Figure 7.** Core allocation type 2: queueing model using 5 cores.

Figure 8 shows the queueing model of core allocation type 3. There are six nodes in the V2X communication device. The core of node 1 is allocated to the packet distributor 1. The cores of nodes 2 and 6 are allocated to packet distributor 2. The core of node 3 is allocated to application 1. The cores of nodes 4 and 5 are allocated to application 2.

In the queueing model of core allocation type 3, the relative arrival rates for $e_i$ are given as follows:

- $e_1 = \alpha$
- $e_2 = (1 - \alpha)/2$

- $e_3 = \alpha\, p_{13} + \{(1 - \alpha)/2\}p_{23} + \{(1 - \alpha)/2\}p_{63} = \beta$
- $e_4 = \alpha\, p_{14} + \{(1 - \alpha)/2\}p_{24} + \{(1 - \alpha)/2\}p_{64} = (1 - \beta)/2$
- $e_5 = \alpha\, p_{15} + \{(1 - \alpha)/2\}p_{25} + \{(1 - \alpha)/2\}p_{65} = (1 - \beta)/2$
- $e_6 = (1 - \alpha)/2$

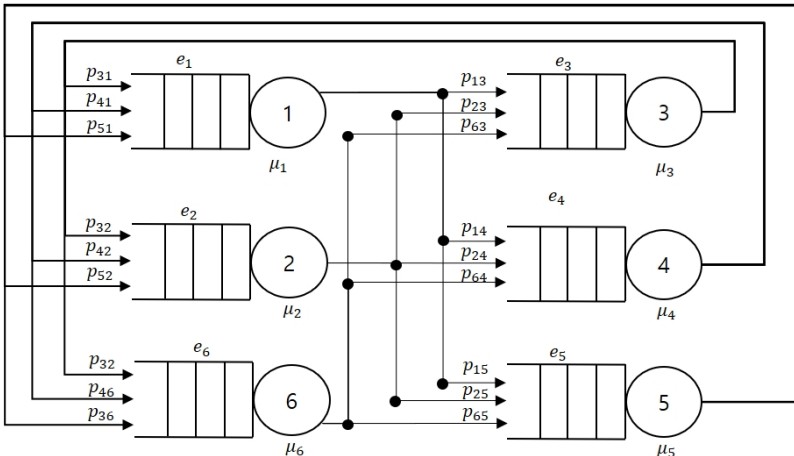

**Figure 8.** Core allocation type 3: queueing model using 6 cores.

The relative arrival rate and routing probability are important factors in program development and performance analysis. Because the relative arrival rate and the routing probability are factors that affect the evaluation result of the V2X communication device to be evaluated, the value should be set, and performance analysis should be performed in consideration of the environment in which the V2X communication device is operated. Therefore, we set $\alpha = 0.1$ so that packets for autonomous/cooperative driving services are more transferred to distributors 2 and application 2.

We assume that the routing probability settings for each type of core allocation are as follows:

- Type 1 (4 cores): $(p_{13}, p_{14}, p_{23}, p_{24}, p_{31}, p_{32}, p_{41}, p_{42}) = (0.9, 0.1, 0.1, 0.9, 0.9, 0.1, 0.1, 0.9)$.
- Type 2 (5 cores): $(p_{13}, p_{14}, p_{15}, p_{23}, p_{24}, p_{25}, p_{31}, p_{32}, p_{41}, p_{42}, p_{51}, p_{52}) = (0.9, 0.05, 0.05, 0.1, 0.45, 0.45, 0.9, 0.1, 0.1, 0.9, 0.1, 0.9)$.
- Type 3 (6 cores): $(p_{13}, p_{14}, p_{15}, p_{23}, p_{24}, p_{25}, p_{31}, p_{32}, p_{36}, p_{41}, p_{42}, p_{46}, p_{51}, p_{52}, p_{55}, p_{63}, p_{64}, p_{65}) = (0.9, 0.05, 0.05, 0.1, 0.45, 0.45, 0.9, 0.05, 0.05, 0.1, 0.45, 0.45, 0.1, 0.45, 0.45, 0.45, 0.45, 0.1)$.

For core allocation type 2, two cores are allocated to application 2, and for core allocation type 3, two cores are allocated to application 2 and packet distributor 2. A packet distribution to the core that executes a similar program code is assumed to operate with a round-robin policy.

*4.2. Result*

Figures 9 and 10 illustrate the performance analysis results for core allocation type 1. Node 3 is allocated as the core for application 1, and node 1 handles most packets for application 1. By contrast, node 4 is allocated as the core for application 2, and node 2 handles most of the packets for application 4.

Figure 9 illustrates throughput versus number of total threads utilizing four nodes in core allocation type 1. It indicates that the overall throughput increased and saturated as the number of threads increased.

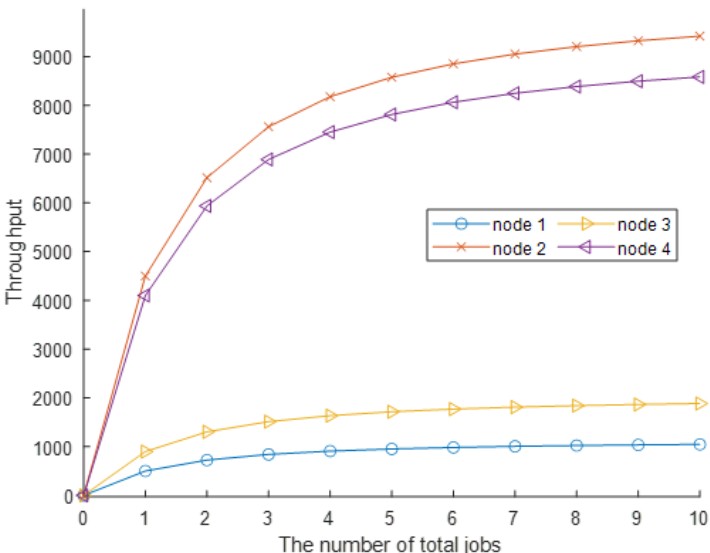

**Figure 9.** Throughput versus number of total threads in core allocation type 1.

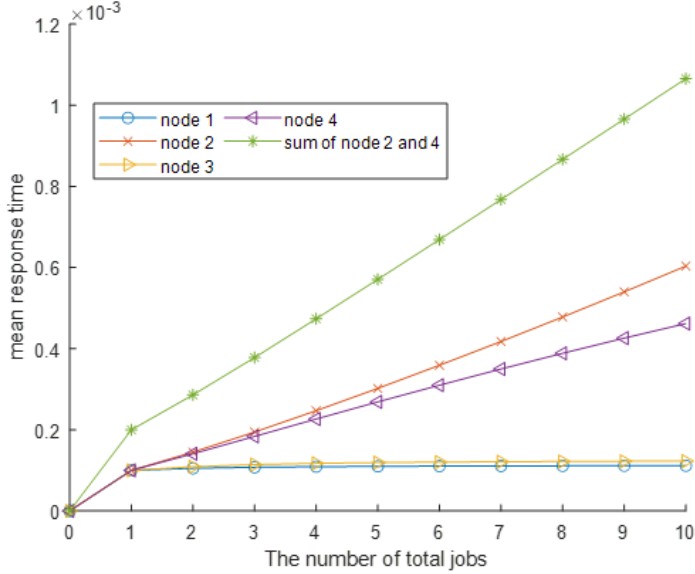

**Figure 10.** Mean response time versus number of total threads in core allocation type 1.

Regarding nodes 1 and 3 for infotainment service, the throughput of node 1 was approximately 1 kpps or less, and the throughput of node 3 was approximately 0.5 kpps.

Regarding nodes 2 and 4 for the autonomous/cooperative driving service, the throughput of node 2 was approximately 9 kpps or less, and the throughput of node 4 was approximately 8.5 kpps or less.

As presented in Table 1, infotainment services required approximately 80 Mbps. This requirement was 156.250 kpps, based on 64-byte packets. Autonomous/cooperative driving services required more than 5 Mbps. This requirement was 9.766 kpps based on 64-byte packets.

However, in core allocation type 1, even if the number of threads increased, the performance requirements could not be satisfied, owing to the limitation of one core. That is, core allocation type 1 required an increase in the number of cores, which was required for performance satisfaction.

Figure 10 illustrates the mean response time versus the number of total threads utilizing four cores in core allocation type 1. The mean response time increased as the number of threads increased.

As presented in Table 1, infotainment services required a latency of less than 1 s. In addition, autonomous/cooperative driving services required a latency of 10 ms or less.

In Figure 10, node 3 illustrates a mean response time of less than 1 s, and node 4 satisfies the performance requirement for a mean response time of 10 ms or less. Because the packet distributor of autonomous/cooperative driving service was mostly handled by packet distributor 2, the latency for the actual autonomous/cooperative driving service could be considered as the sum of the mean response times of nodes 2 and 4. When the mean response time for autonomous/cooperative driving service was considered as the sum of the mean response times of nodes 2 and 4, a performance requirement of 10 ms or less was satisfied.

Both the throughput and the mean response time increased as the number of threads increased. This means that it is better to allocate a significant number of threads to increase throughput, but it is better to allocate fewer threads to reduce latency.

The results of the performance analysis of core allocation type 1 indicate that the latency performance is satisfactory, but the throughput performance is not.

Figures 11 and 12 illustrate the performance analysis results for core allocation type 2. In core allocation type 2, one core is added to application 2 of core allocation type 1. Therefore, application 2 utilizes nodes 4 and 5.

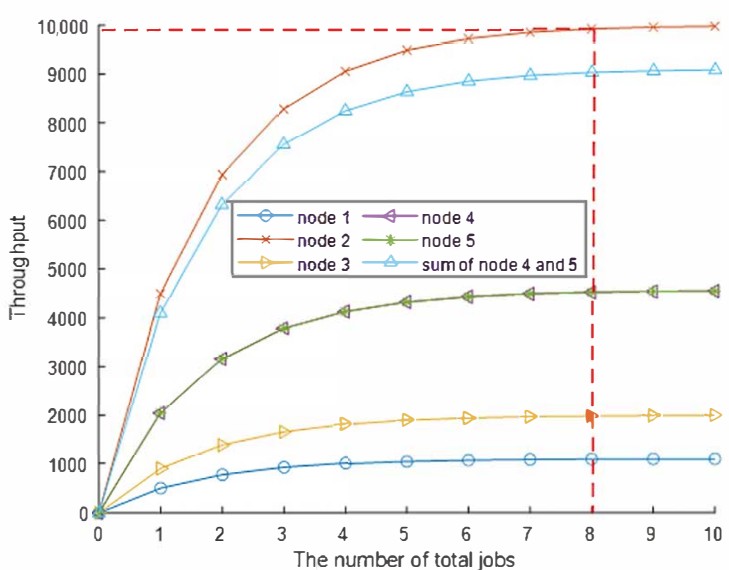

**Figure 11.** Throughput versus number of total threads in core allocation type 2.

Figure 11 illustrates throughput versus number of total threads utilizing five nodes in core allocation type 2. As illustrated in Figure 5, the overall throughput increased and became saturated as the number of threads increased. Compared with Figure 5, because there was no change in the nodes of application 1 and packet distributor 1 of the infotainment service, the throughput of nodes 1 and 3 indicated similar values as in Figure 5. The throughput of node 2 that was allocated for packet distributor 2 of autonomous/cooperative driving services was satisfied when more than eight threads were utilized.

In addition, because the core allocated to application 2 of the autonomous/cooperative driving service increased to nodes 4 and 5, each load of nodes 4 and 5 decreased, but the throughput of application 2, that is, the sum of nodes 4 and 5, increased. However, it can be confirmed that the throughput of the sum of nodes 4 and 5 was approximately 9 kpps, which is less than the throughput requirement of the autonomous/cooperative driving service.

Figure 12 illustrates the mean response time versus the number of total threads utilizing five cores in core allocation type 2. It indicates that the mean response time of core allocation type 2 increased as threads increased.

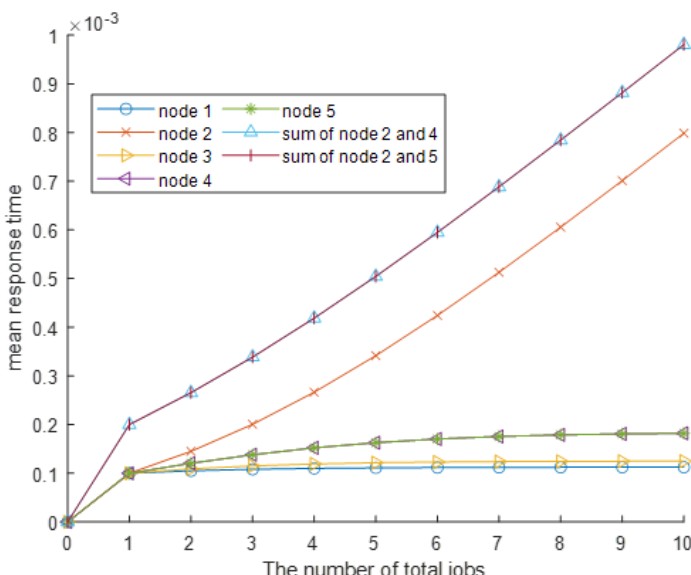

**Figure 12.** Mean response time versus number of total threads in core allocation type 2.

Nodes 1 and 3 satisfy the latency requirement of less than 1 s of infotainment service in less than 0.2 ms. In addition, the mean response time performance of the sum of nodes 2 and 4, and the sum of nodes 2 and 5 for the autonomous/cooperative driving service, satisfied the performance by less than 10 ms.

Comparing the latency of core allocation types 1 and 2, it can be observed that the latency was reduced because the load of application 2 was distributed to nodes 4 and 5 in core allocation type 2. Furthermore, it can be confirmed that the latency of the autonomous/cooperative driving service considering the latency of packet distributor 2 was slightly reduced.

In the performance analysis of core allocation type 2, although it was confirmed that the throughput and mean response time for application 2 increased, compared to core allocation type 1, the throughput requirements for infotainment and autonomous/cooperative driving services were not satisfied.

Figures 13 and 14 illustrate the performance analysis results for core allocation type 3, where one core was added to packet distributor 2 of core allocation type 2. Therefore, packet distributor 2 utilized nodes 2 and 6.

Figure 13 illustrates throughput versus number of total threads utilizing six nodes in core allocation type 3. Like the throughput of core allocation type 1 and type 2, throughput increased and saturated as threads increased. Because the number of cores allocated to application 2 and packet distributor 2 was increased, the respective load on nodes 4 and node 5 decreased, but the throughput of application 2 (sum of nodes 4 and 5) increased.

In particular, the throughput of the sum of nodes 4 and 5, and the sum of nodes 2 and 6, satisfied the throughput requirements for the autonomous/cooperative driving service when four or more threads were allocated. However, it did not satisfy the throughput performance requirements of infotainment services, and for this, it was necessary to add a core for infotainment services.

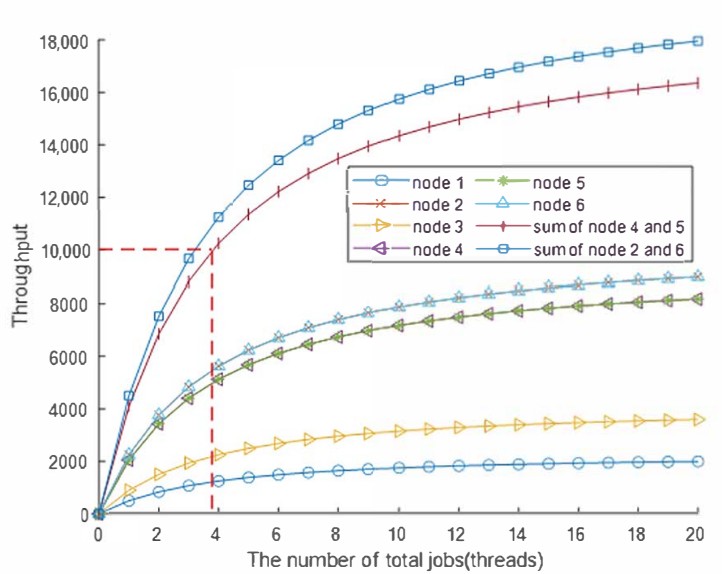

**Figure 13.** Throughput versus number of total threads in core allocation type 3.

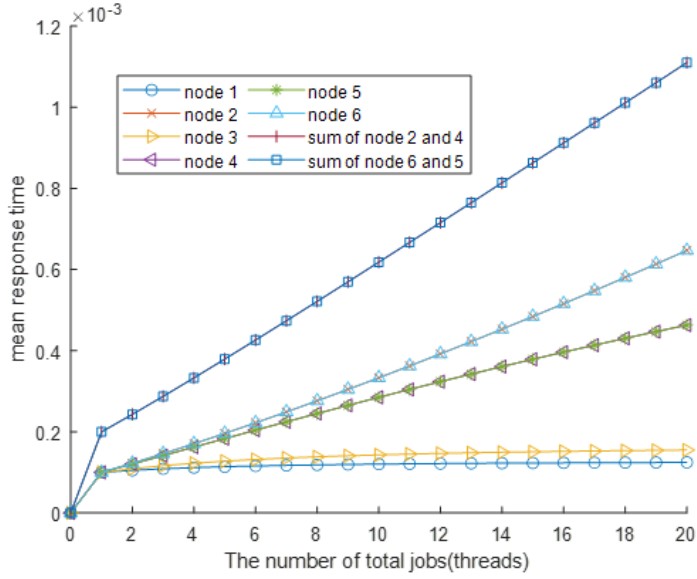

**Figure 14.** Mean response time versus number of total threads in core allocation type 3.

Figure 14 illustrates the mean response time versus number of total threads utilizing six cores in core allocation type 3. Both nodes for the autonomous/cooperative driving and infotainment services had increased latency. However, the latency requirements of 10 ms or less for the autonomous/cooperative driving service, and the latency requirements of 1 s or less for the infotainment service, were satisfied.

Table 3 shows the pros and cons of prototype-level testing and the proposed method. As an advantage of the prototype-level testing, service-, device-, and system-level testing are advantageous for checking performance, including environmental factors that can affect the practical operating environment. Function-related tests (except for performance-related tests) can be performed only through prototype-level testing because they require a developed program. In addition, since this form of testing analyzes the low code at the instruction level, it is possible to accurately find the part causing the performance degradation. However, if it cannot be solved at the instruction level, it may be necessary to return to the design level and re-design the program as well as change the hardware resources. This is a process that consumes significant resources and time. The advantage of

the proposed design-level testing is that it is possible to check the approximate performance expected at the design level, which helps in program planning and hardware selection to achieve performance satisfaction. Therefore, after prototype development, it is possible to prevent the worst situation of having to replace hardware caused by not achieving the performance requirements. However, it is not possible to perform optimized debugging with design-level testing, and the testing method has nothing to do with functional testing.

**Table 3.** Comparison between the prototype-level performance analysis and the proposed method.

| List | Pros | Cons |
| --- | --- | --- |
| The prototype-level performance analysis | • Service-, device-, and system-level testing is advantageous for checking performance, including environmental factors that can affect the practical operating environment.<br>• Function-related tests (except for performance-related tests) are conducted with developed programs, so they can only be performed through prototype-level performance analysis.<br>• Since the instruction-level testing analyzes low codes, the cause of performance degradation can be accurately found. | • If the problem cannot be solved at the instruction level, it may be necessary to return to the design level and re-design the program as well as change the hardware resources.<br>• Complex debugging processes at the instruction level can increase development costs. |
| The proposed method | • The proposed method can confirm the approximate performance of the prototype at the design level, which is helpful in program planning and HW selection to achieve performance satisfaction.<br>• The proposed method can avoid the worst case of having to replace the hardware caused by not achieving the performance requirements. | • After developing the prototype, the developer should eventually conduct instruction-level testing in order to debug for optimization.<br>• It cannot be helpful for functional tests. |

Therefore, it is desirable to perform appropriate hardware selection and algorithm design through performance analysis at the design level in the development process, and to optimize this process through prototype-level performance analysis after developing a prototype product.

**5. Conclusions**

We proposed a modeling and analysis method for CPU resource requirements according to the performance requirements of the V2X communication service programs, and developed a multi-core-based vehicle communication service system in the vehicle network environment.

By modeling and testing the three core allocation types, we confirmed the performance change according to the number of threads and core allocation for the V2X service requirement. We analyzed the core allocation types 1–3 as examples of core resource allocation in order to check the throughput versus the thread and the mean response time with limited resources. In addition, the performance of core allocation types 1–3 was analyzed to ascertain whether the performance was satisfactory for the autonomous/cooperative driving and infotainment applications.

In the case of core allocation type 1, one core was allocated to each of application 1, application 2, packet distributor 1 of WLAN, and packet distributor 2 of Cellular. In the case of core allocation type 2, one core was allocated each to application 1, packet distributor 1 of WLAN, and packet distributor 2 of Cellular, and two cores were allocated to application 2. In the case of core allocation type 3, one core was allocated each to application 1 and packet distributor 1 of WLAN, and two cores were allocated to application 2 and packet distributor 2 of Cellular, respectively.

In this study, it was assumed that application 1 was loaded with a more relevant arrival rate for distributor 1 of WLAN and that application 2 was loaded with a more

relevant arrival rate for distributor 2 of Cellular. Although types 1–3 satisfied the latency performance for vehicle service, more core allocation was required to satisfy the throughput performance requirements. However, both the throughput and mean response time increased as the number of threads increased. To increase throughput, it was better to allocate several threads or cores; to reduce latency, however, it was better to allocate fewer threads and more cores.

Through the analysis, it was confirmed that an appropriate resource allocation algorithm was required to satisfy the performance requirements of various vehicle services with limited resources. Since throughput and latency have a trade-off relationship, there is a limit to infinitely increasing the number of threads. Therefore, it was necessary to construct the design in consideration of the required throughput and latency performance and the number of core resources.

In conclusion, in order to develop a successful V2X communication device that satisfies the performance requirements, appropriate hardware selection and algorithm design should be performed using the proposed method at the design level, and then optimized using the prototype-level performance analysis after prototype development. This method is efficient in terms of development cost and time.

In a further study, we plan to examine the resource distribution or allocation algorithm for performance satisfaction, according to the application service in the V2X communication service system. In addition, using a practical V2X communication program, we plan to compare and analyze the proposed method with the performance results obtained through simulation and implementation of the prototype-level testing method.

**Author Contributions:** Conceptualization, W.-S.C. and S.-G.C.; methodology, W.-S.C.; software, W.-S.C.; validation, W.-S.C. and S.-G.C.; formal analysis, W.-S.C.; investigation, W.-S.C.; resources, W.-S.C.; data curation, W.-S.C.; writing—original draft preparation, W.-S.C.; writing—review and editing, W.-S.C. and S.-G.C.; visualization, W.-S.C.; supervision, S.-G.C.; project administration, S.-G.C.; funding acquisition, S.-G.C. All authors have read and agreed to the published version of the manuscript.

**Funding:** This research was supported by the Basic Science Research Program through the National Research Foundation of Korea (NRF) funded by the Ministry of Education (2020R1A6A1A12047945).

**Institutional Review Board Statement:** Not applicable.

**Informed Consent Statement:** Not applicable.

**Data Availability Statement:** Not applicable.

**Conflicts of Interest:** The authors declare no conflict of interest.

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
