# Peer review of "Thread-Based Modeling and Analysis in Multi-Core-Based V2X Communication Device"

_sustainability, doi:10.3390/su14148277_

Round 1

Reviewer 1 Report

Dear Author ,

The throughput of any node is shown in kbps but most of the time it found Kpps. Please make correction (e.g. line number 531 or line number 491 and so on)

Reviewer 2 Report

In this paper, the authors propose the thread-based modeling and analysis in Vehicle-to-Everything (V2X) communication devices according to the performance requirements of V2X applications, that is utilized to reduce the time and cost of unnecessary work in the development process, and it can be applied to the design level by using the queuing theory. Concretely, they have proposed a thread-level performance modeling and analysis method regarding queuing theory in multi-core-based vehicle service systems and analyzed the performance of multi-core-based vehicle service systems based on the proposed method.

Overall, it is a novel and interesting model, however, some issues still exist in current version. Therefore, I can only suggest publishing this concise paper in sustainability after carefully taking the following suggestions into account in a revision with care and love to detail.

1.    I find some writing errors as well as formatting errors in the paper. For example, the symbol '[ ]' does not match in line 37 and the description of Fig. 3 is redundant in line 379. Besides, the word "where" in Eq. 5 should be lowercase and not indented, and each equation should be followed by a symbol, such as ‘,’ or ‘.’. I suggest that the authors should revise carefully from beginning to end to avoid these errors in detail.

2.   The paper would also be improved if the figure captions would be made more self-contained. In addition to showing what the figures represent, one could also consider using a sentence or two to explain the main message of each figure.

3.    The symbol of each variable should be unique and consistent throughout the full text, while the $\lambda_i$ is present in both Eqs. 12 and 15 but has a different meaning, which may cause ambiguity and the authors should avoid this problem.

4.  The summary of the findings and implications of this paper is insufficient therefore I suggest that the authors should further elaborate their results in the conclusion.

5.    Considering the queueing theory is the main theory in this paper, the following highly related works dealing with queueing models, DOI: 10.1109/TSMC.2022.3149596 and DOI: 10.1109/TSMC.2022.3149596, are recommended to be introduced.

Please also check the English style and spell, I would like to see the revised version soon.

Reviewer 3 Report

The comments are as follows.

1. The problem addressed in the paper is still not clear, especially regarding the practical aspects. For example, in line 89 on page 2, it is difficult to understand why conventional work should perform the debugging process using complex and expensive Instruction-level analysis method for analyzing the performance.

2. In Section 3 "Proposed performance measure of V2X communication device", it is not easy to understand the measure or metrics that define the efficiency performance, such as latency, throughput, and reliability.

3. In section 4.2 Result, the results lack the comparison with the conventional methods.

4. Regarding the performance analysis, the detail setup of the system model, traffic model and application scenarios is not clear. 

5. Categorization of the related work will improve the readability. 

Round 2

Reviewer 2 Report

I agree that the paper can be published, the authors have addressed all my issues.

Author Response

Thank you for your comments.

Reviewer 3 Report

1. It is difficult to understand that the traffic load and practical traffic model are based on practical V2X communication. 

   Furthermore, throughput and latency highly associate with communication performance in addition to queueing performance.  

2. It is still difficult to understand how does the problem addressed in this paper relate with the practical V2X communication.

3. The results lack the comparison with the conventional methods (although it is stated as future study).
